# *Portulaca oleracea* L. Polysaccharide Inhibits Porcine Rotavirus In Vitro

**DOI:** 10.3390/ani13142306

**Published:** 2023-07-14

**Authors:** Xiechen Zhou, Yan Li, Tao Li, Junyang Cao, Zijian Guan, Tianlong Xu, Guiyan Jia, Gaopeng Ma, Rui Zhao

**Affiliations:** 1College of Animal Science and Veterinary Medicine, Heilongjiang Bayi Agricultural University, Daqing 163319, China; zhouxiechen@126.com; 2College of Life Science and Biotechnology, Heilongjiang Bayi Agricultural University, Daqing 163319, China; 15765987571@163.com (Y.L.); gamer_li2023@163.com (T.L.); caojunyang126@163.com (J.C.); byndgzj@163.com (Z.G.); xutianlong451@163.com (T.X.); mgp12318@163.com (G.M.)

**Keywords:** porcine rotavirus, *Portulaca oleracea* L. polysaccharide, antiviral activity

## Abstract

**Simple Summary:**

Porcine rotavirus (PoRV) is a major pathogen causing dehydrating diarrhea and fatality in newborn piglets, resulting in substantial economic losses in the swine industry worldwide. Therefore, the development of a safe and effective medication for treating PoRV infection is imperative. This study provides the first report on the anti-PoRV activity of *Portulaca oleracea* L. (POL). Our findings demonstrate that POL-P has antiviral activity against PoRV infection in vitro, and thus might be developed into a novel antiviral agent to control PoRV in pig farms.

**Abstract:**

Diarrhea is one of the most common causes of death in young piglets. Porcine rotavirus (PoRV) belongs to the genus *Rotavirus* within the family *Reoviridae*, and is considered to be the primary pathogen causing diarrhea in piglets. *Portulaca oleracea* L. (POL) has been reported to alleviate diarrhea and viral infections. However, the antiviral effect of *Portulaca oleracea* L. polysaccharide (POL-P), an active component of POL, on PoRV infection remains unclear. This study demonstrated that the safe concentration range of POL-P in IPEC-J2 cells is 0–400 μg/mL. POL-P (400 μg/mL) effectively inhibits PoRV infection in IPEC-J2 cells, reducing the expression of rotavirus VP6 protein, mRNA and virus titer. Furthermore, on the basis of viral life cycle analysis, we showed that POL-P can decrease the expression of PoRV VP6 protein, mRNA, and virus titer during the internalization and replication stages of PoRV. POL-P exerts antiviral effects by increasing IFN-α expression and decreasing the expression levels of TNF-α, IL-6, and IL-10 inflammatory factors. Overall, our study found that POL-P is a promising candidate for anti-PoRV drugs.

## 1. Introduction

Rotavirus (RV) was first discovered in cattle with diarrhea in 1969 [1]. Porcine rotavirus (PoRV), a member of the genus Rotavirus in the family Reoviridae, is a major cause of dehydrating diarrhea in newborn piglets [2,3]. PoRV diarrhea is an acute and highly contagious disease resulting from PoRV infection [4,5]. PoRV causes severe watery diarrhea, anorexia, vomiting, dehydration, and weight loss in newborn piglets, and the pig industry experiences substantial financial losses due to PoRV [6]. Vaccines are currently commonly used to prevent and control PoRV diarrhea [7,8]. However, owing to the specificity of the vaccine and the presence of multiple genotypes or serotypes of PoRV, the control effects are not satisfactory [9,10,11].

Traditional Chinese medicine (TCM) is a widely accepted, safe, and non-toxic treatment option for various diseases [12]. Numerous studies have demonstrated that the active components in TCM possess antiviral and antimicrobial properties [13,14,15]. For instance, Artemisia annua, an herbal remedy, has been demonstrated to be effective in treating malaria, and Puerarin has been found to inhibit RV replication and proliferation [16,17]. Plant polysaccharides, such as those found in Plantago asiatica L. and seaweed, exhibit therapeutic activities against tumors, inflammation, immunomodulation, and atherosclerosis [18,19,20,21,22]. Specifically, Plantago asiatica L. polysaccharides have been suggested to inhibit pseudorabies virus infection [21]. Recently, research has shown that crude seaweed polysaccharides effectively prevent viral attachment and release in host cells, significantly inhibiting infectious hematopoietic necrosis virus infection [22]. Other studies have demonstrated that Astragalus polysaccharides inhibit spring viremia due to carp virus replication [23].

*Portulaca oleracea* L. (POL) is a common weed in the Portulacaceae family with a global distribution. Recent studies have identified substantial levels of bioactive compounds such as polyphenols, flavonoids, and polysaccharides in POL [24]. Beyond its nutrient-rich composition, POL possesses various pharmacological properties such as anti-inflammatory, anti-tumor, and antiviral effects [25,26,27,28]. Water extracts of POL have been reported to inhibit porcine epidemic diarrhea virus infection in vitro [29] and have been found to effectively alleviate the symptoms of pandemic influenza A virus infection [30]. These findings indicate the potential of POL as a source of natural compounds for the development of novel antiviral agents.

The plant POL has historically been used to treat gastrointestinal ailments and was recorded in the “Compendium of Materia Medica” from the Ming Dynasty. POL has been used in traditional medicine and as a food source in various countries for centuries. In the context of veterinary TCM, POL is considered a possible treatment for clinical diarrhea caused by PoRV. Although previous studies have indicated that POL may possess antiviral properties against different viruses, the antiviral effects of *Portulaca oleracea* L. polysaccharide (POL-P) on PoRV replication remain unclear. This study was aimed at exploring the potential protective effects of POL-P and its ability to inhibit PoRV replication in vitro.

## 2. Materials and Methods

### 2.1. Cells and Virus

The IPEC-J2 cell line was maintained in Dulbecco’s modified Eagle’s medium (DMEM) (Gibco, Grand Island, NE, USA) supplemented with 10% fetal bovine serum (Gibco, Grand Island, NE, USA), and incubated under 5% CO_2_ at 37 °C. The PoRV A strain HJ-2016 (GenBank accession no. MH423866) used in this study was obtained from the Laboratory for the Prevention and Control of Swine Infectious Diseases, College of Animal Science and Veterinary Medicine, Heilongjiang Bayi Agricultural University.

### 2.2. Preparation of POL-P

The POL plant material was procured from Nanjing Shangyuantang pharmacy (Jiangsu, Nanjing, China). POL-P was prepared in our laboratory according to a previously described method [31]. The POL-P was diluted in DMEM to a range of concentrations, including 25, 50, 100, 200, 400, 800, and 1600 μg/mL.

### 2.3. Cell Viability Assays

Cell viability was assessed via CCK-8 assay (Bioshap, Hefei, China) on IPEC-J2 cells. IPEC-J2 cells were seeded at 5 × 10^4^ cells/mL and exposed to various concentrations of POL-P (25–1600.0 μg/mL) for 12, 24, or 48 h. After incubation, cells and 10 μL of CCK-8 reagent were added to each well of a 96-well plate and incubated at 37 °C for 3 h. The absorbance of each sample was measured using a microplate reader (BioTek, Montpelier, VT, USA) at 450 nm. Cell viability percentages were calculated with the following formula: cell viability (%) = [OD (sample) − OD (blank)]/[OD (control) − OD (blank)] × 100%.

### 2.4. Western Blotting

Total cellular proteins were extracted with radioimmunoprecipitation assay buffer (Sigma, St. Louis, MI, USA), separated via 10% SDS-PAGE, and then transferred onto polyvinylidene fluoride membranes. Subsequently, the membranes were blocked for 2 h with 5% nonfat dry milk in phosphate-buffered saline (PBS) containing 0.05% Tween 20 (PBST). Overnight incubation at 4 °C was performed with group A RV, antibodies to VP6 (Biorbyt, Cambridge, UK), and mouse anti-GAPDH antibodies (Abcam, Cambridge, UK). After being washed with 0.05% PBST in Tween 20, the membranes were incubated with HRP-conjugated goat anti-mouse IgG (Proteintech Group, Wuhan, China) secondary antibodies. The membranes were washed with 0.05% PBST and then detected with Luminata Crescendo Western HRP substrate (Merck KGaA, Darmstadt, Germany) on an Amersham Imager 600 (GE Healthcare, Chicago, IL, USA). Finally, the analysis of target protein expression levels was performed in ImageJ 1.8.0 software (Bethesda, Rockville, MD, USA).

### 2.5. RNA Isolation and Real-Time PCR Analysis

Total RNA was extracted from cells with an RNA Simple Total RNA Kit (Tiangen Biotech, Beijing, China). Subsequently, cDNA synthesis was performed with random primers with a FastKing gDNA Dispelling RT SuperMix Kit (Tiangen Biotech, Beijing, China). Absolute quantitative reverse transcription-PCR (qRT-PCR) was conducted with a method described by Su et al. [32] using specific primers (PoRV VP6-F: 5′-GATTCGTGTTCCATAAGCCAAA-3′; PoRV VP6-R: 5′-CTGATCCAGCGTTAATCCACATAG-3′) to generate cDNA via absolute qRT-PCR. The reaction mixture (25 μL) contained 12.5 μL of 2× SYBR Premix Ex Taq (Takara Bio Inc., Kusatsu, Japan), 0.5 μL (10 p mol/L) forward primer, 0.5 μL (10 p mol/L) reverse primer, 4 μL template DNA, and 7.5 μL sterile water. Absolute qRT-PCR was performed with SYBR Green I fluorescent dye and a QuantStudio 3 Real-Time PCR System (Applied Biosystems, Thermo Fisher, Foster, CA, USA). The reaction conditions included denaturation at 95 °C for 30 s; 40 cycles at 95 °C for 30 s, 60 °C for 30 s, and 72 °C for 30 s; and a final dissociation stage. Each sample was assayed three times, and the quantity of PoRV viral RNA was determined on the basis of the PoRV Vp6 standard plasmid results.

### 2.6. TCID_50_ Assays

IPEC-J2 cells were used to determine the viral titer of the PoRV strain HJ-2016 at a multiplicity of infection (MOI) of 1.0. After a 2 h infection period, the cells were washed three times with PBS and the supernatants were removed. Infected cells were refed with DMEM containing 1 mg/mL trypsin. The viral titer of PoRV in the IPEC-J2 cells was determined with the median tissue culture infective dose (TCID_50_). The IPEC-J2 cells were seeded into 96-well plates at a density of 10^5^ cells per well in 100 μL of medium and incubated for 48 h at 37 °C under a 5% CO_2_ atmosphere. Next, 100 μL of 10-fold serial dilutions of virus was added to each well. Cytopathic effects were observed every 12 h for 5 days after inoculation. Finally, the viral titer was calculated according to the Reed–Muench method.

### 2.7. Inhibitory Effects of POL-P at Different Stages of Viral Replication

To investigate the inhibitory effects of POL-P on PoRV replication at different stages of viral infection, we conducted an experiment consisting of five groups, with POL-P (400 μg/mL) and PoRV added at different stages. The first group, the “pretreatment group,” involved pretreating IPEC-J2 cells with POL-P (400 μg/mL) for 6 h before infecting them with PoRV at an MOI of 1.0 for 2 h at 37 °C. Cells were collected at 24 h post-infection (hpi). The second group, the “attachment group,” involved treating cells with POL-P (400 μg/mL) and infecting them with PoRV at an MOI of 1.0 for 2 h at 4 °C. Cells were collected at 24 hpi. The third group, the “internalization group,” involved treating cells with POL-P (400 μg/mL) and infecting them with PoRV at an MOI of 1.0 for 2 h at 37 °C. Cells were collected at 24 hpi. In the fourth group, the “replication group,” cells were infected with PoRV at an MOI of 1.0 for 2 h and then treated with POL-P (400 μg/mL). Cells were collected at 24 hpi. The fifth group, the “full process group,” involved pretreating cells with POL-P (400 μg/mL) for 6 h before infecting them with PoRV at an MOI of 1.0 for 2 h. After infection, cells were treated with POL-P (400 μg/mL) and collected at 24 hpi. The mRNA expression levels of PoRV were analyzed with absolute qRT-PCR.

### 2.8. Indirect Immunofluorescence Assays

The IPEC-J2 cells were fixed in 4% (*w/v*) paraformaldehyde at 4 °C for 1 h, then washed three times with PBS. Next, the cells were permeabilized with 0.2% Triton X-100 for 15 min and blocked with 10% bovine serum albumin (Solarbio, Beijing, China) for 1 h. The cells were then incubated overnight at 4 °C with a solution of antibodies to VP6 (1:2000). After being washed with PBS three times, the cells were incubated in the dark at room temperature for 1 h with FITC-conjugated secondary anti-mouse IgG (1:1000) (ZSGB-BIO, Beijing, China). The cells were then washed three times with PBS, incubated with DAPI (Solarbio, Beijing, China) for 15 min, and washed three times with PBS. The cells were observed under an immunofluorescence microscope (Life Technologies, Foster, CA, USA).

### 2.9. ELISA

Supernatants were collected from IPEC-J2 cells treated at different stages of infection (pretreatment, attachment, internalization, replication, and full process), and levels of cytokines including IFN-α, TNF-α, IL-6, and IL-10 were measured with ELISA kits (Mmbio, Yancheng, China) according to the manufacturer’s instructions. The absorbance at 450 nm was determined using a microplate reader (BioTek, Winooski, VT, USA), and the cytokine concentrations were calculated with a standard curve.

### 2.10. Statistical Analysis

All data are presented as mean ± SD. Statistical analysis was conducted in GraphPad Prism 8.0 software (GraphPad Software, Inc., San Diego, CA, USA). One-way analysis of variance and Student’s *t*-test were used to analyze the data and create the graphs. Statistical significance was defined as *p* < 0.05 (*), *p* < 0.01 (**), and *p* < 0.001 (***).

## 3. Results

### 3.1. Cytotoxicity of POL-P to IPEC-J2 Cells In Vitro

The potential cytotoxicity of POL-P was evaluated with CCK-8 assay to determine the viability of IPEC-J2 cells. The cells were treated with POL-P at various concentrations for 12, 24, and 48 h. After incubation with POL-P at concentrations ranging from 25–400 μg/mL for 12, 24, and 48 h, the viability of IPEC-J2 cells did not significantly differ from that of the control group (0 μg/mL) (*p* > 0.05; Figure 1). These findings revealed that cell viability is not impaired when treated by POL-P at concentration of 25–400 μg/mL. However, at concentrations of 800 and 1600 μg/mL, POL-P significantly decreased cell vitality (*p* < 0.05, *p* < 0.001, respectively; Figure 1). Therefore, the concentrations of POL-P used in subsequent experiments were 100, 200, and 400 μg/mL. Overall, these results suggested that the effective and suitable working concentration of POL-P with IPEC-J2 cells is 25–400 μg/mL.

### 3.2. Different Concentrations of POL-P Inhibit PoRV Infection In Vitro

The levels of VP6 protein, mRNA, and viral titer gradually decreased in PoRV-infected IPEC-J2 cells when treated with POL-P in a dose-dependent manner (100–400 μg/mL). PoRV VP6 protein expression levels decreased approximately 1.19-fold at 24 hpi after treatment with 100 μg/mL POL-P (*p* < 0.05; Figure 2A,B), and PoRV VP6 protein expression levels decreased approximately 1.79-fold and 2.11-fold at 24 hpi after treatment with 200 and 400 μg/mL POL-P, respectively (*p* < 0.001; Figure 2A,B). PoRV mRNA levels decreased by approximately 1.22-fold and 1.17-fold, respectively, compared with DMEM, and concentrations of 200 and 400 μg/mL POL-P decreased the PoRV mRNA levels by approximately 1.32-fold and 1.45-fold, respectively, compared with DMEM, at 24 hpi (*p* < 0.001; Figure 2C). TCID_50_ assays indicated that the viral titer in the PoRV cells (4.19 ± 0.09 log_10_ TCID_50_/mL) was significantly higher than that in the 200 and 400 μg/mL POL-P treated cells (3.88 ± 0.07 and 3.82 ± 0.05 log_10_ TCID50/mL, respectively; *p <* 0.001; Figure 2D). These data demonstrate that POL-P inhibits PoRV replication in vitro.

### 3.3. Effects of POL-P Inhibition on PoRV Life Cycle Stages

To determine the effects of POL-P on PoRV pretreatment, attachment, internalization, replication, and the full process, we detected the levels of PoRV mRNA and VP6 protein by absolute qRT-PCR, western blotting, and indirect immunofluorescence assays. During the internalization, replication stage, and full process, POL-P decreased PoRV mRNA levels by approximately 1.22-fold, 1.17-fold, and 1.47-fold, respectively, compared with control groups (*p* < 0.01; Figure 3). POL-P had no effect on PoRV pretreatment and attachment in IPEC-J2 cells (*p* > 0.05; Figure 3). Compared with the control groups, the levels of VP6 protein were lower in PoRV-infected IPEC-J2 cells, and the VP6 protein expression levels decreased approximately 2.11-fold, 2.36-fold, and 3.16-fold for internalization, replication, and the full process, respectively, after treatment with 400 μg/mL POL-P (*p* < 0.001; Figure 4A,B). Moreover, indirect immunofluorescence assays revealed that VP6 protein expression was lower after viral infection than in the control groups (Figure 4C). The above data confirmed that POL-P inhibits PoRV by interfering with the internalization and replication stages of PoRV.

### 3.4. Effect of POL-P on Cytokine Release in IPEC-J2 during Different Processes

Using the ELISA method, we detected inflammatory factors secreted by IPEC-J2 cells. Compared with those in the PoRV group, during the internalization process, POL-P significantly decreased the expression levels of TNF-α, IL-6, and IL-10 cytokines, with inhibition rates of 32.32%, 22.58%, and 28.25%, respectively (*p* < 0.001, Figure 5A–C), and increased IFN-α expression by approximately 1.13-fold (*p* < 0.001, Figure 5D). Compared with the PoRV group, during the PoRV replication stage, POL-P treatment inhibited the production of TNF-α, IL-6, and IL-10 inflammatory factors in cells, with inhibition rates of 39.99%, 28.95%, and 36.54%, respectively (*p* < 0.01, *p* < 0.001, Figure 5A,C,D), and significantly increased IFN-α expression by approximately 1.08-fold (*p* < 0.01, Figure 5B).

## 4. Discussion

Worldwide, PoRV leads to large economic losses in the swine industry. PoRV is transmitted primarily through a fecal-oral route. When it invades the body, it settles mainly in the small intestines of pigs, causing damage to the intestinal villi, resulting in acute diarrhea, and ultimately leading to acute dehydration and death of piglets. A safe and effective drug is needed to treat PoRV infection. POL-P was previously synthesized by our laboratory and purified from purslane. The molecular weight of the homogeneous polysaccharide POL-P is 4 × 10^4^ Da; it was purified to a concentration greater than 95% and was found to consist of seven monosaccharides linked by β-glycosidic bonds. The content comprises 33.5% glucuronic acid (GlcA), 32.2% galactose, 15.4% arabinose, 13.2% rhamnose, 3.3% glucose, 1.2% mannose, and 1.2% galacturonic acid [31]. TCM has the advantage of being multi-component, multi-target, and multi-pathway [33,34]. In this study, POL-P was found to inhibit PoRV infection in vitro and to aid in controlling PoRV infection in pig farms.

Polysaccharides in TCM are natural polymers formed from the aggregation of different monosaccharides via hydrogen bonds or van der Waals forces. They offer benefits of safety, low toxicity, and broad biological activity [35]. However, different concentrations of polysaccharides have certain effects on cells [36]. The safe concentration range of Glycyrrhiza polysaccharide in PK15 cells is 0–600 μg/mL, while the safe concentration range of Huaier polysaccharide e in PK15 cells is 0–200 μg/mL [37,38]. The safe concentration range of POL-P for IPEC-J2 cells was determined to be 0–400 μg/mL using a CCK8 assay. Structural proteins have long been considered important targets for antigen detection, and VP6 of RV is a conserved protein [39,40]. Dose-dependent effects of POL-P on PoRV VP6 protein, mRNA, and viral titer were observed. POL-P at a concentration of 400 μg/mL had the greatest inhibitory effect on PoRV. Consequently, choosing an optimal concentration of POL-P is crucial [41]. Viral infection of cells involves multiple processes. The process through which POL-P inhibits PoRV is currently unclear. Our findings demonstrated, on the basis of analysis of the levels of VP6, that POL-P strongly inhibits PoRV during the internalization and replication processes. Similar results have been observed in the inhibition of viral infection by other polysaccharides. Aloe extract exhibits the strongest antiviral effects in the replication process of porcine epidemic diarrhea virus [42]. However, Taishan Pinus massoniana pollen polysaccharides have the strongest inhibitory effect on H9N2 subtype avian influenza virus at the time of viral invasion [43]. Glycyrrhiza polysaccharide exhibits potent inhibitory effects during the attachment, adhesion, and internalization stages of PRV [37].

RV infects mature enterocytes at the villus tip in the jejunum and ileum, causing diarrhea and destruction of these cells [44]. Early laboratory studies indicated that POL-P has the potential to treat ulcerative-colitis-associated diarrhea. Studies have shown that different polysaccharides have inhibitory effects on different viruses in vivo. Crude polysaccharides from seaweed and abalone viscera have antiviral activity against SARS-CoV-2 [45]. Polysaccharides from Thais clavigera (Küster) show significant anti-hepatitis B virus activity by enhancing immune cell function [46]. Isatis indigotica polysaccharide effectively inhibits the expression of IP-10, IL-6, MIG, and CCL-5 induced by human influenza virus (PR8/H1N1) and avian influenza virus (H9N2) [47]. Infection with porcine circovirus 2 (PCV2) and pseudorabies virus (PRV) can enhance the expression of IFN-α in the host [48]. The significant expression of IFN-α plays a crucial role in the antiviral response of the organism. IL-6, IL-10, and TGF are cytokines associated with inflammation. Overexpression of these cytokines suggests a state of inflammation within the body, while the return of their expression to a normal level indicates a reduction in the body’s inflammatory state. POL-P exerts antiviral effects by regulating the immune status of the host, thereby increasing IFN-α expression and decreasing the expression levels of TNF-α, IL-6, and IL-10 inflammatory factors. Overall, results suggest that these polysaccharides are promising candidates for antiviral drugs.

The mechanism of disease resistance of polysaccharides is complex. POL-P is gradually decomposed into soluble sugar under the action of enzymes, with each playing a role at its respective position. GlcA has the highest content among constituent carbohydrates of POL-P, and studies have shown that GlcA has anti-viral effects. Sulfated GlcA may be the main structure inhibiting the binding of SARS-CoV-2 to host cells [49]. Heterosubtypic protection of infected cells has been found to be induced by a live attenuated influenza virus vaccine with galactose-α-1,3-galactose epitopes [50].

Although we confirmed that POL-P inhibits PoRV in vitro, the exact underlying mechanism remains unclear. Further studies are expected to contribute to the development of effective antiviral drugs for preventing PoRV infection.

## 5. Conclusions

In conclusion, this study is the first report on the anti-PoRV activity of POL-P. Our findings revealed that POL-P has antiviral activity against PoRV infection in vitro and thus might be developed into a novel antiviral agent to control PoRV in pig farms.

## Figures and Tables

**Figure 1 animals-13-02306-f001:**
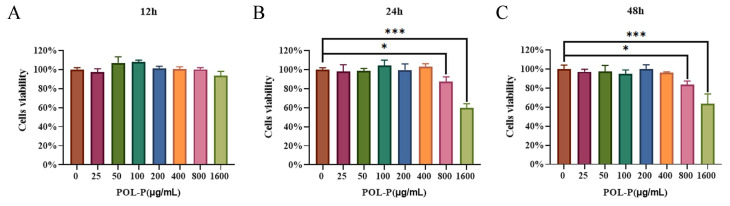
Cytotoxicity of POL-P in IPEC-J2 cells: (**A**) effects of different concentrations (0–1600 μg/mL) of POL-P on the viability of IPEC-J2 cells after 12 h of treatment; (**B**) after 24 h of treatment; (**C**) after 48 h of treatment. Data are presented as mean ± SD (*n* = 3). *, *p* < 0.05 and ***, *p* < 0.001.

**Figure 2 animals-13-02306-f002:**
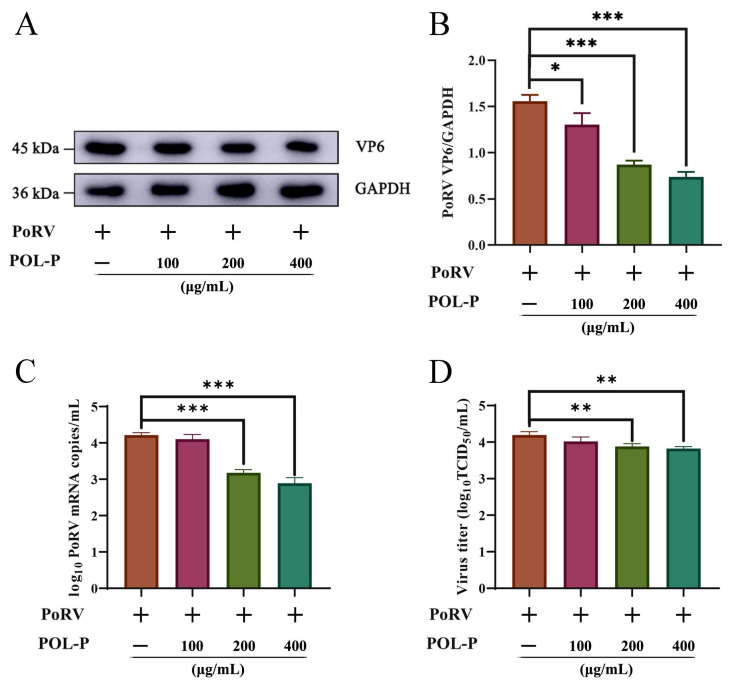
Different concentrations of POL-P inhibit PoRV infection. (**A**) Immunoblot analysis of VP6 protein levels in IPEC-J2 cells infected with PoRV and treated with different concentrations of POL-P (100, 200, and 400 μg/mL) for 24 h. (**B**) The expression levels of VP6 protein were normalized to those of GADPH. (**C**) IPEC-J2 cells were treated with different concentrations of POL-P (100, 200, and 400 μg/mL) and infected with PoRV for 24 h. Subsequently, absolute qRT-PCR assays were performed to analyze the collected cells. (**D**) PoRV viral titers were measured in IPEC-J2 cells treated with different concentrations of POL-P (100, 200, and 400 μg/mL) for 24 h. The cells and supernatants were collected for TCID50 tests. Data are presented as mean ± SD (*n* = 3). *, *p* < 0.05; **, *p* < 0.01, and ***, *p* < 0.001.

**Figure 3 animals-13-02306-f003:**
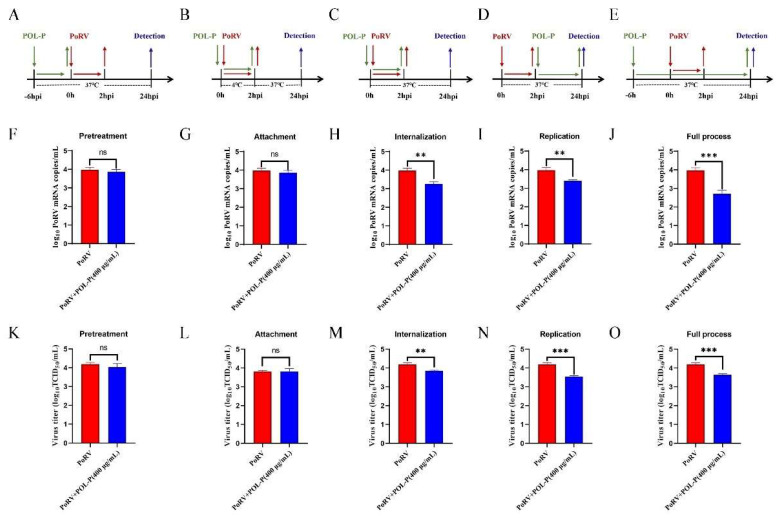
Effects of POL-P on mRNA and viral titers of PoRV during pretreatment, attachment, internalization, replication, and the full process. (**A**–**E**) Pattern diagrams of pretreatment, attachment, internalization, replication, and the full process. (**F**–**J**) Detection of PoRV mRNA during pretreatment, attachment, internalization, replication, and the full process. (**K**–**O**) Detection of viral titers of PoRV during pretreatment, attachment, internalization, replication, and the full process. Data are presented as mean ± SD (*n* = 3). **, *p* < 0.01, and ***, *p* < 0.001.

**Figure 4 animals-13-02306-f004:**
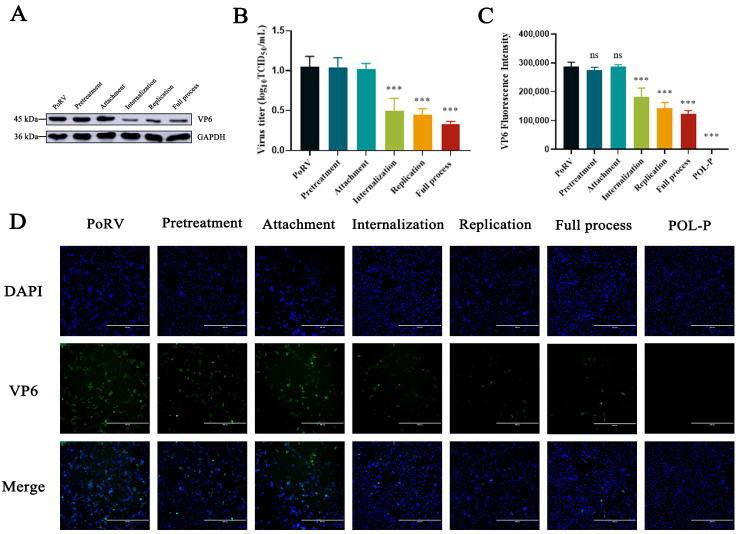
Effects of POL-P on VP6 protein of PoRV during pretreatment, attachment, internalization, replication, and the full process. (**A**) Immunoblot analysis of the VP6 protein levels in IPEC-J2 cells infected with PoRV and treated with POL-P (400 μg/mL) during pretreatment, attachment, internalization, replication, and the full process. (**B**) Levels of VP6 fluorescence protein expression analyzed in ImageJ, normalized to that of GADPH. (**C**) Intensity of VP6 protein expression analyzed in Image J. (**D**) Localization of PoRV VP6 protein (green) in IPEC-J2 cells, visualized with PoRV immunofluorescence during pretreatment, attachment, internalization, replication, and the full process (scale bars = 400 μm). Data are presented as mean ± SD (*n* = 3). ***, *p* < 0.001.

**Figure 5 animals-13-02306-f005:**
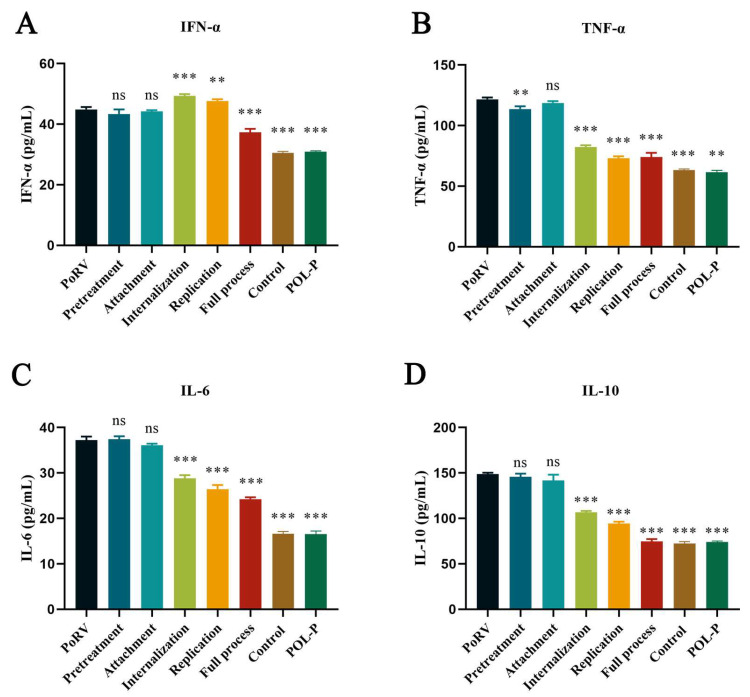
Effects of POL-P on cytokine release by IPEC-J2 during pretreatment, attachment, internalization, replication, and the full process. (**A**) Effects of POL-P (400 μg/mL) on TNF-α in IPEC-J2 cells infected with PoRV during pretreatment, attachment, internalization, replication, and the full process. (**B**) Effects of POL-P (400 μg/mL) on INF-α of IPEC-J2 cells infected with PoRV during pretreatment, attachment, internalization, replication, and the full process. (**C**) Effects of POL-P (400 μg/mL) on IL-6 of IPEC-J2 cells infected with PoRV during pretreatment, attachment, internalization, replication, and the full process. (**D**) Effects of POL-P (400 μg/mL) on IL-10 of IPEC-J2 cells infected with PoRV during pretreatment, attachment, internalization, replication, and the full process. Data are presented as mean ± SD (*n* = 3). **, *p* < 0.01, and ***, *p* < 0.001.

## Data Availability

The data that support the findings of this study are available from the corresponding author (R.Z.) upon reasonable request.

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
