# Peer review of "Portulaca oleracea L. Polysaccharide Inhibits Porcine Rotavirus In Vitro"

_animals, 2023, doi:10.3390/ani13142306_

Round 1

Reviewer 1 Report

The manuscript is intresting and is helpful for understanding the antiviral effects of purslane polysaccharides. But the research design and data results need to be supplemented and improved, and the writing of discussion sections, abstracts, etc. also needs to be revised as following.

1. In cells experiments on the effects of purslane polysaccharides on virus infection and cytokine expressions, purslane polysaccharide treatment controls need be added.

2. The image of IFA for virus detection are blurry (the pixels should be too low); Additionally, statistical analysis of fluorescence intensity in IFA results should be added.

3. The author used a polysaccharide extraction solution with a dose of 400ug/mL, which was relatively high in concentration and close to the safe dose for cytotoxicity. Based on the IFA results (too few cells in the pretreatment group, adhesion group, and internalization group), it was deduced that the dose could cause significant toxic effect on the cells.

4. In abstract, necessary data results should be added.

5. In discussions, necessary comparative analysis of data results such as inhibition of purslane polysaccharides on viral infection process, cytokines production, and safe doses of cytotoxicity should be more supplied.

Author Response

Response to Reviewer 1 Comments

Point 1: In cells experiments on the effects of purslane polysaccharides on virus infection and cytokine expressions, purslane polysaccharide treatment controls need be added.

Response 1: Thanks for reviewer's suggestions. In the cell experiment examining the effects of purslane polysaccharides (POL-P) on cytokine expression, a POL-P treatment group was added, and the figures were redrawn accordingly. In virus infection experiments, the detection of the target virus was used as an indicator [1, 2].However, the addition of POL-P did not enable the detection of the virus, and therefore, the POL-P group was not included as a supplement in this part of the experiment. The ”PoRV+POL-P (Full process)” group involved the addition of POL-P throughout the entire process of virus infection, while the PoRV group only involved the process of virus infection without POL-P. Comparing the two groups allows for an evaluation of the role of POL-P in the process of virus infection.

Point 2: The image of IFA for virus detection are blurry (the pixels should be too low); Additionally, statistical analysis of fluorescence intensity in IFA results should be added.

Response 2: Thanks for reviewer's suggestions.The image of IFA has been redrawn, and the fluorescence intensity was calculated by Image J software and the Figure 4 were redrawn accordingly. The low clarity of the image may be due to a reduction in pixel count that occurred when the image was inserted into Word. To address this issue, the IFA images have been uploaded as attachments to ensure their quality.

Point 3: The author used a polysaccharide extraction solution with a dose of 400ug/mL, which was relatively high in concentration and close to the safe dose for cytotoxicity. Based on the IFA results (too few cells in the pretreatment group, adhesion group, and internalization group), it was deduced that the dose could cause significant toxic effect on the cells.

Response 3: Thanks for reviewer's suggestions. The results in Figure 1 indicate that incubation of IPEC-J2 cells with 400 μg/mL of POL-P for 48 hours did not affect cell viability, as compared to the blank control group. The selection of 400 μg/mL of POL-P was based on the results shown in Figure 2, which demonstrated a dose-dependent inhibition of the virus by POL-P. Therefore, the maximum dose of POL-P within the safe concentration range was selected for subsequent experiments, and a concentration of 400 μg/mL was used. The low number of cells in the previous images (Figure 4) may have been due to the low total cell count and the random nature of the field of view. To address this issue, the IFA experiment was repeated, and the images were redrawn accordingly.

Point 4: In abstract, necessary data results should be added.

Response 4: Thanks for reviewer's suggestions. The abstract was rewritten as follows:

Diarrhea is one of the most important causes of death in young piglets. Porcine rotavirus (PoRV) belong to the genus rotavirus within the family Reoviridae, are considered as the most important pathogens that cause diarrhea in piglets. Portulaca oleracea L. (POL) has been reported to alleviate diarrhea and viral infections. However, the antiviral effect of Portulaca oleracea L. polysaccharide (POL-P), an active component of POL, on PoRV infection remains unclear. This study demonstrated that the safe concentration range of POL-P on IPEC-J2 cells is 0-400 μg/mL. POL-P (400 μg/mL) effectively inhibits PoRV infection in IPEC-J2 cells, reducing the expression of rotavirus VP6 protein, mRNA and virus titer. And POL-P can decrease the expression of PoRV VP6 protein, mRNA, and virus titer during the internalization and replication stages of PoRV, on the basis of viral life cycle analysis. POL-P exerts antiviral effects by increasing IFN-α expression, and decreasing the expression levels of TNF-α, IL-6, and IL-10 inflammatory factors. Overall, our study find that POL-P are promising candidates for anti-PoRV drugs.

Point 5: In discussions, necessary comparative analysis of data results such as inhibition of purslane polysaccharides on viral infection process, cytokines production, and safe doses of cytotoxicity should be more supplied.

Response 5: Thanks for reviewer's suggestions. The discussions was rewritten as follows:

Worldwide, PoRV leads to large economic losses in the swine industry. PoRV is transmitted primarily through a fecal-oral route. When it invades the body, it settles mainly in the small intestine of pigs, thus causing damage to the intestinal villi, resulting in acute diarrhea, and ultimately leading to acute dehydration and death of piglets. A safe and effective drug is needed to treat PoRV infection. POL-P was previously synthesized by our laboratory and purified from purslane. The molecular weight of the homogeneous polysaccharide POL-P is 4 × 104 Da; it was purified to a concentration greater than 95% and was found to consist of seven monosaccharides linked by β-glycosidic bonds. The content comprises 33.5% glucuronic acid (GlcA), 32.2% galactose, 15.4% arabinose, 13.2% rhamnose, 3.3% glucose, 1.2% mannose, and 1.2% galacturonic acid [3]. TCM has the advantage of being multi-component, multi-target, and multi-pathways [4, 5]. In this study, POL-P was found to inhibit PoRV infection in vitro and to aid in controlling PoRV infection in pig farms.

Polysaccharides in TCM are natural polymers formed from the aggregation of different monosaccharides via hydrogen bonds or van der Waals forces. They offer benefits of safety, low toxicity, and broad biological activity [6]. However, different concentrations of polysaccharides have certain effects on cells [7]. The safe concentration range of Glycyrrhiza Polysaccharide in PK15 cells is 0-600 μg /mL, while the safe concentration range of Huaier Polysaccharide e in PK15 cells is 0-200 μg/mL [8, 9]. The safe concentration range of POL-P for IPEC-J2 cells was determined to be 0–400 μg/mL with CCK8 assay. Structural proteins have long been considered important targets for antigen detection, and VP6 of RV is a conserved protein [10, 11]. Dose-dependent effects of POL-P on PoRV VP6 protein and mRNA, and viral titer were observed. POL-P at a concentration of 400 μg/mL had the greatest inhibitory effect on PoRV. Consequently, choosing an optimal concentration of POL-P is crucial [12]. Viral infection of cells involves multiple processes. The process through which POL-P inhibits PoRV is currently unclear. Our findings demonstrated that POL-P strongly inhibits PoRV during the internalization and replication processes, on the basis of analysis of the levels of VP6. Similar results have been observed in the inhibition of viral infection by other polysaccharides. Aloe extract exhibits the strongest antiviral effects in the replication process of porcine epidemic diarrhea virus [1]. However, Taishan Pinus massoniana pollen polysaccharides have the strongest inhibitory effect on H9N2 subtype avian influenza virus at the time of viral invasion [13]. Glycyrrhiza Polysaccharide exhibit potent inhibitory effects during the attachment, adhesion, and internalization stages of PRV [8].

RV infects mature enterocytes at the villus tip in the jejunum and ileum, thus causing diarrhea and destruction of these cells [14]. Early laboratory studies indicated that POL-P has the potential to treat ulcerative colitis-associated diarrhea. Studies have shown that different polysaccharides have inhibitory effects on different viruses in vivo. Crude polysaccharides from seaweed and abalone viscera have antiviral activity against SARS-CoV-2 [15]. Polysaccharides from Thais clavigera (Küster) show significant anti-hepatitis B virus activity by enhancing immune cell function [16]. Isatis indigotica polysaccharide effectively inhibits the expression of IP-10, IL-6, MIG, and CCL-5 induced by human influenza virus (PR8/H1N1) and avian influenza virus (H9N2) [17]. Infection with Porcine circovirus 2 (PCV2) and pseudorabies virus (PRV) can enhance the expression of IFN-α in the host [18]. The significant expression of IFN-α plays a crucial role in the antiviral response of the organism. IL-6, IL-10, and TGF are cytokines associated with inflammation. Over expression of these cytokines suggests a state of inflammation within the body, while returning their expression to a normal level indicates a reduction in the body's inflammatory state. POL-P exerts antiviral effects by regulating the immune status of the host, thereby increasing IFN-α expression, and decreasing the expression levels of TNF-α, IL-6, and IL-10 inflammatory factors. Overall, results suggest that these polysaccharides are promising candidates for antiviral drugs.

The mechanism of disease resistance of polysaccharides is complex. POL-P is gradually decomposed into soluble sugar under the action of enzymes. Each plays a role at its respective position. GlcA has the highest content among constituent carbohydrates of POL-P, and studies have shown that GlcA has anti-viral effects. Sulfated GlcA may be the main structure inhibiting the binding of SARS-CoV-2 to host cells [19]. Heterosubtypic protection of infected cells has been found to be induced by a live attenuated influenza virus vaccine with galactose-α-1,3-galactose epitopes [20].

Although we confirmed that POL-P inhibits PoRV in vitro, the exact underlying mechanism remains unclear. Further studies are expected to contribute to the development of effective antiviral drugs to prevent PoRV infection.

References

  1. Xu, Z.; Liu, Y.; Peng, P.; Liu, Y.; Huang, M.; Ma, Y.; Xue, C.; Cao, Y., Aloe extract inhibits porcine epidemic diarrhea virus in vitro and in vivo. Vet Microbiol 2020, 249, 108849.
  2. Liu, Y.; Zhao, L.; Xie, Y.; Chen, Z.; Yang, S.; Yin, B.; Li, G.; Guo, H.; Lin, S.; Wu, J., Antiviral activity of portulaca oleracea L. extracts against porcine epidemic diarrhea virus by partial suppression on myd88/NF-kappab activation in vitro. Microb Pathog 2021, 154, 104832.
  3. Jia, G.; Shao, X.; Zhao, R.; Zhang, T.; Zhou, X.; Yang, Y.; Li, T.; Chen, Z.; Liu, Y., Portulaca oleracea L. polysaccharides enhance the immune efficacy of dendritic cell vaccine for breast cancer. Food Funct 2021, 12, (9), 4046-4059.
  4. Gao, L.; Hao, J.; Niu, Y. Y.; Tian, M.; Yang, X.; Zhu, C. H.; Ding, X. L.; Liu, X. H.; Zhang, H. R.; Liu, C.; Qin, X. M.; Wu, X. Z., Network pharmacology dissection of multiscale mechanisms of herbal medicines in stage IV gastric adenocarcinoma treatment. Medicine (Baltimore) 2016, 95, (35), e4389.
  5. Liu, X.; Wu, J.; Zhang, D.; Wang, K.; Duan, X.; Zhang, X., A Network Pharmacology Approach to Uncover the Multiple Mechanisms of Hedyotis diffusa Willd. on Colorectal Cancer. Evid Based Complement Alternat Med 2018, 2018, 6517034.
  6. Puluhulawa, L. E.; Joni, I. M.; Mohammed, A. F. A.; Arima, H.; Wathoni, N., The Use of Megamolecular Polysaccharide Sacran in Food and Biomedical Applications. Molecules 2021, 26, (11).
  7. Du, S.; Han, B.; Li, K.; Zhang, X.; Sha, X.; Gao, L., Lycium barbarum Polysaccharides Protect Rat Corneal Epithelial Cells against Ultraviolet B-Induced Apoptosis by Attenuating the Mitochondrial Pathway and Inhibiting JNK Phosphorylation. Biomed Res Int 2017, 2017, 5806832.
  8. Huan, C.; Xu, Y.; Zhang, W.; Ni, B.; Gao, S., Glycyrrhiza Polysaccharide Inhibits Pseudorabies Virus Infection by Interfering with Virus Attachment and Internalization. Viruses 2022, 14, (8).
  9. Huan, C.; Yao, J.; Xu, W.; Zhang, W.; Zhou, Z.; Pan, H.; Gao, S., Huaier Polysaccharide Interrupts PRV Infection via Reducing Virus Adsorption and Entry. Viruses 2022, 14, (4).
  10. Yan, X. Y.; Wang, Y.; Xiong, L. F.; Jian, J. C.; Wu, Z. H., Phylogenetic analysis of newly isolated grass carp reovirus. Springerplus 2014, 3, 190.
  11. James, V. L.; Lambden, P. R.; Caul, E. O.; Clarke, I. N., Enzyme-linked immunosorbent assay based on recombinant human group C rotavirus inner capsid protein (VP6) To detect human group C rotaviruses in fecal samples. J Clin Microbiol 1998, 36, (11), 3178-81.
  12. Gaffney, K. J.; Urban, T. A.; Lucena, M.; Anwer, F.; Dean, R. M.; Gerds, A. T.; Hamilton, B. K.; Jagadeesh, D.; Kalaycio, M. E.; Khouri, J.; Pohlman, B.; Sobecks, R.; Winter, A.; Rybicki, L.; Majhail, N. S.; Hill, B. T., Toxicity analysis of busulfan pharmacokinetic therapeutic dose monitoring. J Oncol Pharm Pract 2022, 10781552221104422.
  13. Shang, H.; Sha, Z.; Wang, H.; Miao, Y.; Niu, X.; Chen, R.; Huang, J.; Huang, H.; Wei, K.; Zhu, R., Taishan Pinus massoniana pollen polysaccharide inhibits H9N2 subtype influenza virus infection both in vitro and in vivo. Vet Microbiol 2020, 248, 108803.
  14. Chepngeno, J.; Takanashi, S.; Diaz, A.; Michael, H.; Paim, F. C.; Rahe, M. C.; Hayes, J. R.; Baker, C.; Marthaler, D.; Saif, L. J.; Vlasova, A. N., Comparative Sequence Analysis of Historic and Current Porcine Rotavirus C Strains and Their Pathogenesis in 3-Day-Old and 3-Week-Old Piglets. Front Microbiol 2020, 11, 780.
  15. Yim, S. K.; Kim, K.; Kim, I. H.; Chun, S. H.; Oh, T. H.; Kim, J. U.; Kim, J. W.; Jung, W. H.; Moon, H. S.; Ku, B. S.; Jung, K. J., Inhibition of SARS-CoV-2 Virus Entry by the Crude Polysaccharides of Seaweeds and Abalone Viscera In Vitro. Mar Drugs 2021, 19, (4).
  16. Tang, F.; Huang, G.; Lin, L.; Yin, H.; Shao, L.; Xu, R.; Cui, X., Anti-HBV Activities of Polysaccharides from Thais clavigera (Kuster) by In Vitro and In Vivo Study. Mar Drugs 2021, 19, (4).
  17. Li, Z.; Li, L.; Zhou, H.; Zeng, L.; Chen, T.; Chen, Q.; Zhou, B.; Wang, Y.; Chen, Q.; Hu, P.; Yang, Z., Radix isatidis Polysaccharides Inhibit Influenza a Virus and Influenza A Virus-Induced Inflammation via Suppression of Host TLR3 Signaling In Vitro. Molecules 2017, 22, (1).
  18. Li, X.; Chen, S.; Zhang, L.; Niu, G.; Zhang, X.; Yang, L.; Ji, W.; Ren, L., Coinfection of Porcine Circovirus 2 and Pseudorabies Virus Enhances Immunosuppression and Inflammation through NF-kappaB, JAK/STAT, MAPK, and NLRP3 Pathways. Int J Mol Sci 2022, 23, (8).
  19. Xu, Y.; Li, Y.; You, X.; Pei, C.; Wang, Z.; Jiao, S.; Zhao, X.; Lin, X.; Lu, Y.; Jin, C.; Gao, G. F.; Li, J.; Wang, Q.; Du, Y., Novel Insights Into the Sulfated Glucuronic Acid-Based Anti-SARS-CoV-2 Mechanism of Exopolysaccharides From Halophilic Archaeon Haloarcula hispanica. Front Chem 2022, 10, 871509.
  20. Yan, L. M.; Lau, S. P. N.; Poh, C. M.; Chan, V. S. F.; Chan, M. C. W.; Peiris, M.; Poon, L. L. M., Heterosubtypic Protection Induced by a Live Attenuated Influenza Virus Vaccine Expressing Galactose-alpha-1,3-Galactose Epitopes in Infected Cells. mBio 2020, 11, (2).

Reviewer 2 Report

The manuscript by Zhou and colleagues reports the results regarding the effects of the polysaccharide of the weed Portulaca oleracea L. (POL-P) used in the traditional Chinese medicine on the infection by porcine rotavirus (PoRV), one of the main viruses involved in neonatal diarrhea in piglets and associated mortality.

The study is interesting and well developed as experimental design, and the materials and methods are properly used. The results support the effects of this compound in inhibiting or interfering rotavirus infection in a porcine intestinal cell line (IPEC-J2) widely used for in vitro studies.

POL-P proved to be efficacious in preventing or reducing the efficiency of some phases of PoRV infection such as internalization and replication, as well as reducing the IPEC-J2 release of TNF-alpha, IL-6, and IL-10, and inducing IFN-alpha release.

The discussion is well approached even if it could be strengthened in some parts, providing more detailed comments to the results.

In my opinion, the article can be evaluated positively for publication after revision regarding the presentation of some results in order to provide a better readability and clarify some aspects in the results (cytokine ELISA) and related discussion.

Also minor points should be addressed. 

Specific comments on the text: 

Simple summary

Line 4 specify that you are referring to “PoRV infection”

Line 5 please provide some examples or details about where this effect (to alleviate diarrhea and viral infections) has been observed

Line 6 what do you mean with “remain unclear”? please reword or specify better

Abstract

Lines 1-6 the first lines are identical to the first lines in the simple summary, please change them or reword

Lines 8-10 also these last lines are identical, please reword

Introduction

Page 1 line 8 I’d use another term instead of polygenotypes (e.g. multiple genotypes)

Page 2 line 18 please specify what you are referring to with “POL extracts to inhibit porcine diarrhea virus”. Page 2 line 29-30 what do you mean with “remain uncertain”? please specify. What do you mean with “radioprotective”? please specify in the text or reword

Materials and methods

Cells and virus

Line 1 usually the medium for IPEC-J2 culture is supplemented with F12 medium, why did you use DMEM only?

Cell viability assays

Line 1 did you use one CCK-8 assay or multiple different assays? It is not clear. If only one, use singular

Line 3 did you use trypsin to collect 100 ul of IPEC-J2 after incubation?

Line 5 “by using a microplate reader”

RNA isolation and real-Time PCR analysis

Last lines please specify what plasmid you used for standard

Par 2.7 inhibitory effects

Line 3 “at different stages”

Par 2.9 ELISA

Line 1 “at different stages”

Line 3-4 “by using ELISA kits” please include details of the kits and the plate reader

Results

3.1

Line 1 “by a CCK-8 assay” (if one)

Line 2, 4 what do you mean with “relative” and “rate” related to viability? Specify or omit

Line 6 I would not use the term “biocompatible” as this term usually refers to materials used with biological matrices, not molecules

Line 7-8 please fuse the two sentences, as they are repetitive

Line 10 I’d omit “limited to”

Line 11 I would change “good biocompatibility” with another term

Fig 1

The font size is too small, it not easy to read, please magnify

Can you explain why you use a y-axis higher that 100%? What do you refer to with this percentage of viability? Can you clarify?

You could avoid repeating “ug/ml” in all x-axis values

Caption: line 1 “in IPEC-J2”. I suggest fusing the sentences, and not repeating the whole sentence for each condition

Page 5, 3.2

Line 1-2 the first lines are not clear, please specify the dose of what you used

Line 6 “PoRV”

Line 7 use “fold”, not “times”

Last line “demonstrate”

Fig 2

The font size is too small, please magnify. You can omit “ug/ml” in the x-axis designations

3.3

Line 3 “by absolute qRT-PCR”

Line 4-6 please specify when (in which stage) these values were recorded, or if they were recorded in the full process

Fig 3

The font size is too small, please magnify. I suggest using a horizontal pagination and bigger writings

Fig 4A The font size is too small, please magnify.

3.4

Title I would indicate “cytokine release”

Line 1 I’d say “secreted by IPEC-J2”

Line 3 IL-10 is not an inflammatory factor as it shuts off inflammation. So rephrase please

Line 4 I’d use “cytokines” instead of “factors”

Lines 5, 6, 9, 10 the references to fig 5 letters is not correct.

Line 6 “replication stage”

Fig 5

The font size is too small, please magnify.

Caption.

Please fuse the multiple sentences in only one sentence, as the caption is very repetitive for each cytokine

Line 1 “on cytokine release by”

Last line (and whole experiments) does “n = 3” mean that you performed 3 independent experiments? Please specify in the materials and methods

Discussion

Page 9

Line 3 “small intestine”

Line 5 I suggest making a less obvious statement, so please rephrase or reword: “A safe… treat PoRV”

Lines  6-7 the sentence is not very clear. Can you specify? “The homogeneous… 104 Da”

Line 11 “TCM has the advantage of being …”

Last line is it “CCK8 assays”? plural or singular?

Page 10

Lines 7-8 I’d use “demonstrated” instead of “verified”. “the levels of VP6”

Lines 23-25 there is no proper discussion on the modulation of cytokine release, and in addition IL-10 is not inflammatory. Please discuss the cytokine response more and report/discuss IL-10 result separately. Did you analyze IL-10 or IL-12?

Figure 2 raw data immunoblotting. Why did you report stimulation of IPEC-J2 for 48 h? this is not present in the materials and methods.

Data availability statement. Insert a statement according to your data.

References

7. title. Rotavirus vaccines…

The English language is fine. Only minor corrections must be addressed.

Author Response

Response to Reviewer 2 Comments

Point 1: The manuscript by Zhou and colleagues reports the results regarding the effects of the polysaccharide of the weed Portulaca oleracea L. (POL-P) used in the traditional Chinese medicine on the infection by porcine rotavirus (PoRV), one of the main viruses involved in neonatal diarrhea in piglets and associated mortality.The study is interesting and well developed as experimental design, and the materials and methods are properly used. The results support the effects of this compound in inhibiting or interfering rotavirus infection in a porcine intestinal cell line (IPEC-J2) widely used for in vitro studies.POL-P proved to be efficacious in preventing or reducing the efficiency of some phases of PoRV infection such as internalization and replication, as well as reducing the IPEC-J2 release of TNF-alpha, IL-6, and IL-10, and inducing IFN-alpha release. The discussion is well approached even if it could be strengthened in some parts, providing more detailed comments to the results.In my opinion, the article can be evaluated positively for publication after revision regarding the presentation of some results in order to provide a better readability and clarify some aspects in the results (cytokine ELISA) and related discussion. Also minor points should be addressed.

Specific comments on the text:

Response 1: Thanks for reviewer's suggestions. The specific modifications of the manuscript are as follows.

Point 2: Simple summary

Line 4 specify that you are referring to “PoRV infection”

Response 2: Thanks for reviewer's suggestions. Simple summary line 4: “PoRV” was replaced by “PoRV infection”.

Point 3: Line 5 please provide some examples or details about where this effect (to alleviate diarrhea and viral infections) has been observed.

Response 3:

Thanks for reviewer's suggestions. Simple summary line 5: This description “to alleviate diarrhea and viral infections” does not fit “simple summary” and has been removed.

Point 4: Line 6 what do you mean with “remain unclear”? please reword or specify better

Response 4:

Thanks for reviewer's suggestions. Simple summary line 6: This description “remain unclear” does not fit “simple summary” and has been removed.

Point 5: Abstract

Lines 1-6 the first lines are identical to the first lines in the simple summary, please change them or reword

Lines 8-10 also these last lines are identical, please reword

Response 5: Thanks for reviewer's suggestions. The manuscript had been revised according to the reviewer ’s suggestion.

Point 6: Introduction

Page 1 line 8 I’d use another term instead of polygenotypes (e.g. multiple genotypes)

Response 6: Thanks for reviewer’s suggestion. The “polygenotypes” was replaced by “multiple genotypes”

Point 7: Page 2 line 18 please specify what you are referring to with “POL extracts to inhibit porcine diarrhea virus”.

Response 7: Thanks for reviewer’s suggestion. The “POL extracts to inhibit porcine diarrhea virus” can be replaced by “Water extract of POL has been reported to inhibit porcine epidemic diarrhea virus infection in vitro”

Point 8: Page 2 line 29-30 what do you mean with “remain uncertain”? please specify. What do you mean with “radioprotective”? please specify in the text or reword

Response 8: Thanks for reviewer’s suggestion. The “remain uncertain” means whether POL-P has an antiviral effect on porcine rotavirus. “the effects of Portulaca oleracea L. polysaccharide (POL-P) on PoRV replication remain uncertain.” is replaced by “the antiviral effects of Portulaca oleracea L. polysaccharide (POL-P) on PoRV replication remain unclear”. “radioprotective” is modified to “protective”. The manuscript had been revised.

Point 9: Materials and methods

Cells and virus

Line 1 usually the medium for IPEC-J2 culture is supplemented with F12 medium, why did you use DMEM only?

Response 9: Thanks for reviewer’s suggestion. IPEC-J2 cells in our laboratory maintained well cell morphology after prolonged passage in DMEM medium, so we used DMEM for cell culture.

Point 10: Cell viability assays

Line 1 did you use one CCK-8 assay or multiple different assays? It is not clear. If only one, use singular

Response 10: Thanks for reviewer’s suggestion. The manuscript had been revised according to the reviewer ’s suggestion. “CCK-8 assays” is modified to “CCK-8 assay”.

Point 11: Line 3 did you use trypsin to collect 100 ul of IPEC-J2 after incubation?

Response 11: Thanks for reviewer’s suggestion. The manuscript was not clearly expressed and has been revised.

Point 12: Line 5 “by using a microplate reader”

Response 12: Thanks for reviewer’s suggestion. The manuscript had been revised according to the reviewer ’s suggestion. “ with microplate spectrophotometer” is modified to “by using a microplate reader”.

Point 13: RNA isolation and real-Time PCR analysis

Last lines please specify what plasmid you used for standard

Response 13: Thanks for reviewer’s suggestion. The manuscript had been revised according to the reviewer ’s suggestion.

Point 14: Par 2.7 inhibitory effects

Line 3 “at different stages”

Response 14: Thanks for reviewer’s suggestion. The manuscript had been revised according to the reviewer ’s suggestion.

Point 15: Par 2.9 ELISA

Line 1 “at different stages”

Response 15: Thanks for reviewer’s suggestion. The manuscript had been revised according to the reviewer ’s suggestion.

Point 16: Line 3-4 “by using ELISA kits” please include details of the kits and the plate reader

Response 16: Thanks for reviewer’s suggestion. The manuscript had been revised according to the reviewer ’s suggestion.

Point 17: Results

3.1

Line 1 “by a CCK-8 assay” (if one)

Response 17: Thanks for reviewer’s suggestion. The manuscript had been revised according to the reviewer ’s suggestion.

Point 18: Line 2, 4 what do you mean with “relative” and “rate” related to viability? Specify or omit

Response 18: Thanks for reviewer’s suggestion. The manuscript had been revised according to the reviewer ’s suggestion. “relative viability rate” is modified to “viability”.

Point 19: Line 6 I would not use the term “biocompatible” as this term usually refers to materials used with biological matrices, not molecules

Response 19: Thanks for reviewer’s suggestion. This description “biocompatible” does not fit. The manuscript had been revised to “These findings revealed that cell viability is not impaired when treated by POL-P at concentration of 25–400 μg/mL”

Point 20: Line 7-8 please fuse the two sentences, as they are repetitive

Response 20: Thanks for reviewer’s suggestion. The manuscript had been revised according to the reviewer ’s suggestion.

Point 21: Line 10 I’d omit “limited to”

Response 21: Thanks for reviewer’s suggestion. The manuscript had been revised according to the reviewer ’s suggestion.

Point 22: Line 11 I would change “good biocompatibility” with another term

Response 22: Thanks for reviewer’s suggestion. The manuscript had been revised according to the reviewer ’s suggestion.

Point 23: Fig 1

The font size is too small, it not easy to read, please magnify

Can you explain why you use a y-axis higher that 100%? What do you refer to with this percentage of viability? Can you clarify?

You could avoid repeating “ug/ml” in all x-axis values

Caption: line 1 “in IPEC-J2”. I suggest fusing the sentences, and not repeating the whole sentence for each condition

Response 23: Thanks for reviewer’s suggestion. The figure and caption had been revised according to the reviewer ’s suggestion. Cells viability refers to the ratio of viable cells to the control group after co-incubation of POL-P and IPEC-J2 cells. Cells viability of the 50 μg/mL POL-P group was slightly higher than that of the control group, but the difference was not statistically significant. This may be attributed to the ability of low concentrations of POL-P to reduce cell apoptosis. The small font size in the images may be due to a reduction in pixel count that occurred when the images were inserted into Word. To address this issue, all images have been uploaded as attachments to ensure their quality.

Point 24: Page 5, 3.2

Line 1-2 the first lines are not clear, please specify the dose of what you used

Response 24: Thanks for reviewer’s suggestion. The manuscript had been revised according to the reviewer ’s suggestion.

Point 25: Line 6 “PoRV”

Response 25: Thanks for reviewer’s suggestion. The manuscript had been revised according to the reviewer ’s suggestion.

Point 26: Line 7 use “fold”, not “times”

Response 26: Thanks for reviewer’s suggestion. The manuscript had been revised according to the reviewer ’s suggestion.

Point 27: Last line “demonstrate”

Response 27: Thanks for reviewer’s suggestion. The manuscript had been revised according to the reviewer ’s suggestion.

Point 28: Fig 2

The font size is too small, please magnify. You can omit “ug/ml” in the x-axis designations

Response 28: Thanks for reviewer’s suggestion. The figure had been revised according to the reviewer ’s suggestion.

Point 29: 3.3

Line 3 “by absolute qRT-PCR”

Response 29: Thanks for reviewer’s suggestion. The manuscript had been revised according to the reviewer ’s suggestion.

Point 30: Line 4-6 please specify when (in which stage) these values were recorded, or if they were recorded in the full process

Response 30: Thanks for reviewer’s suggestion. The manuscript had been revised according to the reviewer ’s suggestion.

Point 31: Fig 3

The font size is too small, please magnify. I suggest using a horizontal pagination and bigger writings

Fig 4A The font size is too small, please magnify.

Response 31: Thanks for reviewer’s suggestion. The font of the image has been enlarged.

Point 32: Title I would indicate “cytokine release”

Response 32: Thanks for reviewer’s suggestion. The manuscript had been revised according to the reviewer ’s suggestion.

Point 33: Line 1 I’d say “secreted by IPEC-J2”

Response 33: Thanks for reviewer’s suggestion. The manuscript had been revised according to the reviewer ’s suggestion.

Point 34: Line 3 IL-10 is not an inflammatory factor as it shuts off inflammation. So rephrase please

Response 34: Thanks for reviewer’s suggestion. The manuscript had been revised according to the reviewer ’s suggestion. “inflammation factors” is modified to “cytokines”.

Point 35: Line 4 I’d use “cytokines” instead of “factors”

Response 35: Thanks for reviewer’s suggestion. The manuscript had been revised according to the reviewer ’s suggestion.

Point 36: Lines 5, 6, 9, 10 the references to fig 5 letters is not correct.

Response 36: Thanks for reviewer’s suggestion. The manuscript had been revised according to the reviewer ’s suggestion.

Point 37: Line 6 “replication stage”

Response 37: Thanks for reviewer’s suggestion. The manuscript had been revised according to the reviewer ’s suggestion.

Point 38: Fig 5

The font size is too small, please magnify.

Response 38: Thanks for reviewer’s suggestion. The figure had been revised according to the reviewer ’s suggestion.

Point 39: Caption.

Please fuse the multiple sentences in only one sentence, as the caption is very repetitive for each cytokine

Line 1 “on cytokine release by”

Last line (and whole experiments) does “n = 3” mean that you performed 3 independent experiments? Please specify in the materials and methods

Response 39: Thanks for reviewer’s suggestion. The figure and caption had been revised according to the reviewer ’s suggestion.

Point 40: Discussion

Page 9

Line 3 “small intestine”

Response 40: Thanks for reviewer’s suggestion. The manuscript had been revised according to the reviewer ’s suggestion.

Point 41: Line 5 I suggest making a less obvious statement, so please rephrase or reword: “A safe… treat PoRV”

Response 41: Thanks for reviewer’s suggestion. The manuscript had been revised according to the reviewer ’s suggestion.

Point 42: Lines  6-7 the sentence is not very clear. Can you specify? “The homogeneous… 104 Da”

Response 42: Thanks for reviewer’s suggestion. The sentence is modified to “the homogeneous polysaccharide POL-P is 4 × 104 Da”.

Point 43: Line 11 “TCM has the advantage of being …”

Response 43: Thanks for reviewer’s suggestion. The manuscript had been revised according to the reviewer ’s suggestion.

Point 44: Last line is it “CCK8 assays”? plural or singular?

Response 44: Thanks for reviewer’s suggestion. The manuscript had been revised according to the reviewer ’s suggestion. The “CCK8 assays” is modified to “CCK8 assay”.

Point 45: Page 10

Lines 7-8 I’d use “demonstrated” instead of “verified”. “the levels of VP6”

Response 45: Thanks for reviewer’s suggestion. The manuscript had been revised according to the reviewer ’s suggestion.

Point 46: Lines 23-25 there is no proper discussion on the modulation of cytokine release, and in addition IL-10 is not inflammatory. Please discuss the cytokine response more and report/discuss IL-10 result separately. Did you analyze IL-10 or IL-12?

Response 46: Thanks for reviewer’s suggestion. The manuscript had been revised according to the reviewer ’s suggestion.

Point 47: Figure 2 raw data immunoblotting. Why did you report stimulation of IPEC-J2 for 48 h? this is not present in the materials and methods.

Data availability statement. Insert a statement according to your data.

Response 47: Thanks for reviewer’s suggestion. The manuscript had been revised according to the reviewer ’s suggestion. Figure 1 indicates that incubation with POL-P for 48 hours did not affect cell viability in IPEC-2 cells. In Figure 2, cell samples were tested 24 hours after virus inoculation to prevent cell detachment caused by Cytopathic Effect.

Point 48: References

  1. title. Rotavirus vaccines…

Response 48: Thanks for reviewer’s suggestion. The manuscript had been revised according to the reviewer ’s suggestion.
